Endosymbiont and gut bacterial communities of the brown-banded cockroach, Supella longipalpa

http://orcid.org/0000-0001-8508-5960 Guse Kylene
Pietri Jose E. Jose.Pietri@usd.edu
Division of Basic Biomedical Sciences, University of South Dakota , Vermillion, SD , United States
Sunny Armando
Electronic publication date: 2024 Mar 19
Publication date: 2024
Volume: 12
Electronic Location ID: e17095
Received 2023 Dec 22; Accepted 2024 Feb 21
Copyright: © 2024 Guse and Pietri
Copyright year: 2024
Copyright holder: Guse and Pietri
License: This is an open access article distributed under the terms of the Creative Commons Attribution License, which permits unrestricted use, distribution, reproduction and adaptation in any medium and for any purpose provided that it is properly attributed. For attribution, the original author(s), title, publication source (PeerJ) and either DOI or URL of the article must be cited.
License URL: https://creativecommons.org/licenses/by/4.0/

Keywords: Brown-banded cockroach, Supella longipalpa, Microbiome, Microbiota, 16S, Endosymbiont, Wolbachia, Blattabacterium

Funding: National Institutes of Health, National Institute of Allergy and Infectious Diseases R01AI171014 This work was funded by the National Institutes of Health, National Institute of Allergy and Infectious Diseases grant R01AI171014 to Jose Pietri. The funders had no role in study design, data collection and analysis, decision to publish, or preparation of the manuscript.

==============================
The brown-banded cockroach (Supella longipalpa) is a widespread nuisance and public health pest. Like the German cockroach (Blattella germanica), this species is adapted to the indoor biome and completes the entirety of its life cycle in human-built structures. Recently, understanding the contributions of commensal and symbiotic microbes to the biology of cockroach pests, as well as the applications of targeting these microbes for pest control, have garnered significant scientific interest. However, relative to B. germanica, the biology of S. longipalpa, including its microbial associations, is understudied. Therefore, the goal of the present study was to quantitatively examine and characterize both the endosymbiont and gut bacterial communities of S. longipalpa for the first time. To do so, bacterial 16S rRNA gene amplicon sequencing was conducted on DNA extracts from whole adult females and males, early instar nymphs, and late instar nymphs. The results demonstrate that the gut microbiome is dominated by two genera of bacteria known to have beneficial probiotic effects in other organisms, namely Lactobacillus and Akkermansia. Furthermore, our data show a significant effect of nymphal development on diversity and variation in the gut microbiome. Lastly, we reveal significant negative correlations between the two intracellular endosymbionts, Blattabacterium and Wolbachia, as well as between Blattabacterium and the gut microbiome, suggesting that Blattabacterium endosymbionts could directly or indirectly influence the composition of other bacterial populations. These findings have implications for understanding the adaptation of S. longipalpa to the indoor biome, its divergence from other indoor cockroach pest species such as B. germanica, the development of novel control approaches that target the microbiome, and fundamental insect-microbe interactions more broadly.

Introduction

The brown-banded cockroach, Supella longipalpa, is an invasive nuisance and public health pest native to northern Africa that infests built environments across all inhabited continents. Its general biology has been reviewed in detail relatively recently (Nasirian, 2016) and therefore will only be briefly discussed here. Like its more prevalent relative, the German cockroach (Blattella germanica), S. longipalpa is an omnivorous species that completes its life cycle strictly indoors. However, S. longipalpa differs from B. germanica in numerous aspects of its physiology and behavior, including nutrient selection (Cohen et al., 1987), thermal preference (Tsai & Chi, 2007), and oviposition patterns (Benson & Huber, 1989), among others. Moreover, relative to B. germanica, fundamental biological studies of S. longipalpa are lacking.

In recent years, understanding the roles of commensal and symbiotic microbes in the biology of pest cockroaches has garnered significant interest across scientific disciplines (Latorre et al., 2022; Pan, Wang & Zhang, 2020), and the microbiomes of over 19 different omnivorous species have been sequenced (Tinker & Ottesen, 2020). The microbiome of the German cockroach (Kakumanu et al., 2018) has been the most extensively characterized and studied, revealing its involvement in olfaction and dietary preference (Pérez-Cobas et al., 2015; Zhu et al., 2022), aggregation behavior (Wada-Katsumata et al., 2015), susceptibility to insecticides (Pietri, Tiffany & Liang, 2018; Wolfe & Scharf, 2022), and colonization resistance against ingested human pathogenic bacteria (Ray, Potts & Pietri, 2020; Turner, Van Hulzen & Pietri, 2023). These findings have highlighted the potential applications of targeting the microbial communities of cockroaches for pest control and disease prevention, spurring additional work investigating the use of antimicrobials to effectively manipulate said communities (Li et al., 2020; Rosas et al., 2018; Wolfe & Scharf, 2021; Zha et al., 2023).

In contrast to B. germanica, little is known about the microbial communities associated with S. longipalpa. Several reports have described the isolation of diverse Gram-positive and Gram-negative bacteria from field-collected S. longipalpa (Nasirian, 2016). For example, Citrobacter spp., Enterobacter spp., Serratia spp., Klebsiella spp., and Pseudomonas spp. have all been successfully cultured from S. longipalpa (Le Guyader, Rivault & Chaperon, 1989). However, the abundances of these microbes and whether they represent contaminants or stable associations are unclear. Further, as culture independent high throughput sequencing methods have not been used to assess the microbial communities of S. longipalpa, the vast majority of taxa have likely been overlooked.

An additional intriguing microbial aspect of S. longipalpa is its relationship with the vertically transmitted insect endosymbiont Wolbachia. Most cockroaches are known to be colonized by Blattabacterium, a cockroach specific, vertically transmitted endosymbiont (Noda et al., 2020), but not by Wolbachia. On the other hand, S. longipalpa is one of few cockroach species that has been shown to be colonized by Wolbachia at a high prevalence, in addition to Blattabacterium (Choubdar et al., 2023; Gibson & Hunter, 2009; Oladipupo et al., 2023; Vaishampayan et al., 2007). Yet, the relative balance of the two endosymbionts throughout development, and their possible connections to bacteria in the gut, are unknown.

To add to the scarce body of knowledge of microbial interactions in S. longipalpa, in the current study we sought to quantitatively examine and characterize both its endosymbiont and gut bacterial communities. To do so, bacterial 16S rRNA gene amplicon sequencing was conducted on DNA extracts from whole adult females and males, early instar nymphs, and late instar nymphs. We then analyzed community composition and differences in relative taxonomic abundances and diversity indices across the sexes and developmental stages.

Materials and Methods

Cockroaches

A colony of the KSU strain of S. longipalpa (Roach Crossing, Royal Oak, MI, USA) was maintained in a plastic enclosure at the University of South Dakota insectary facility on a 12:12 photoperiod at ~35% relative humidity. The cockroaches were provided dog chow as a food source (Purina, St. Louis, MO, USA) as well as tap water and cardboard harborages for shelter. Six adult males, six adult females, six early instar nymphs, and six late instar nymphs were collected directly from the colony for analysis. Adults were not specifically aged, and instars were separated based solely on size. S. longipalpa typically undergoes 6–8 molts (Tsai & Chi, 2007), and early instars and late instars in our study corresponded to 1st/2nd instars (very small) and likely >5th instars of nearly adult size, respectively. Each insect was washed in solutions of 10% bleach and 70% ethanol to remove surface contaminants before being allowed to dry and stored at –20 °C until further processing.

DNA extraction, sequencing and data processing

DNA was isolated from individual frozen cockroaches using a DNeasy blood and tissue kit (Qiagen, Germantown, MD, USA) according to the manufacturer’s protocol. DNA concentration was then measured using a Qubit fluorometer (Thermo Fisher Scientific, Waltham, MA, USA). Primers targeting the V4 hypervariable region of the bacterial 16S rRNA gene (515/806) were used for PCR with HotStarTaq Plus Master Mix (Qiagen, Hilden, Germany). 95 °C, 53 °C, and 72 °C were used as denaturation, annealing, and extension temperatures, respectively. PCR products then underwent electrophoresis on 2% agarose gel to verify successful amplification. Samples were multiplexed using unique dual indices and pooled together at equal concentrations before being purified using calibrated Ampure XP beads (Beckman Coulter, Brea, CA, USA). Pooled and purified PCR products were used as input for Illumina DNA library preparation. Sequencing was performed at MR DNA (Shallowater, TX, USA) on an Illumina MiSeq instrument (Illumina, San Diego, CA, USA). A mock extraction control was also performed and sequenced.

16S rRNA sequences were processed using the Qiime2 pipeline (qiime2.org). In short, raw sequencing data were processed to remove primers and low-quality reads (phred quality score < 25). High-quality reads were considered for denoising, merging and chimera removal, and to generate unique amplicon sequence variants (ASVs) using the Dada2 plugin within Qiime2 (Bolyen et al., 2019). Representative sequences of each ASV were aligned using MAFFT and phylogenetic trees both rooted and unrooted were constructed with FastTree (Price, Dehal & Arkin, 2009). Taxonomic assignments of bacterial ASVs were based on reference sequences (clustered at 99% sequence identity) from the Silva 132 reference database (Release 128).

Sequencing depth ranged from 114,154 to 291,026 with a mean of 211,494 sequences and a SEM of ± 9,394. A mock extraction sample (blank control) was sequenced and used to assess for any contamination patterns in experimental samples before downstream analysis. To normalize the data, ASVs which were not present in at least three samples were omitted, and the bacterial ASV table produced was converted to relative proportions using total reads per sample. Microbial community analyses were performed both with and without endosymbiont reads to consider the gut and endosymbiont communities separately and in relation to each other. For analyses conducted without the endosymbiont reads (Blattabacterium and Wolbachia), columns of reads assigned to these taxa were deleted from tables before relative abundance transformation, completely removing them from the analyses while leaving the gut bacterial community intact.

Data analysis

All analyses were performed in the R statistical platform version 4.3.1 (R Core Team, 2023). Briefly, for alpha diversity, beta diversity and permutational multivariate analyses of variance (PERMANOVA), multiple R packages such as vegan and ape were used (Oksanen et al., 2020; Paradis, Claude & Strimmer, 2004). CLR transformation was done on the genus table to estimate the relative proportions using the robCompositions package (Templ, Hron & Filzmoser, 2011). Any significant outliers were detected using the dixon.test in the Outliers package in R (Komsta, 2022) and were removed from the analysis. The shapiro.test was used in R to determine normality and the appropriate statistical tests were performed. A one-way ANOVA was used to test statistical significance of differentially abundant bacteria. Kruskal–Wallis tests were used to check the statistical significance among multiple groups using the kruskal.test function, whereas Wilcoxon rank-sum tests were used to determine the statistical significance of inter-individual variations between each developmental stage/sex. Correlations were performed with the cor function from the ‘stats’ package in R. Heatmaps were generated using heatmap3 function (Zhao et al., 2021) and the correlelogram was generated using the corrplot function (version 0.92) in R. All other visualizations were plotted using ggplot2 function (Wickham, Chang & Wickham, 2016) in R.

Results

Taxonomic signatures among nymph and adult S. longipalpa

Taxonomic assessment of the most abundant bacterial families among S. longipalpa identified Blattabactericeae (59.3%), Anaplasmataceae (24.9%), Desulfovibrionaceae (2.5%), Akkermansiaceae (2.2%), Bacteroidaceae (1.7%), Lactobacillaceae (1.6%) and Tannerellaceae (1.3%), among a few others (Fig. 1A). As Blattabacteriaceae and Anaplasmataceae are endosymbionts of S. longipalpa that do not reside in the gut, we performed analysis without these taxa, which identified Lactobacillaceae (21.6%), Desulfovibrionaceae (13.9%), Akkermensiaceae (13.5%), Bacteroidaceae (7.8%), Rikenellaceae (7.7%), Tannerellaceae (6.2%) and Ruminococceae (4.9%) as the most abundant bacterial families (Fig. 1B).

Figure 1 Taxonomic composition of the microbiome of S. longipalpa at the family and genus levels.

(A) Relative abundances of the top 20 most abundant families with the endosymbionts included. (B) Relative abundances of the top 20 most abundant families without the endosymbionts included. (C) Relative abundances of the top 20 most abundant genera with the endosymbionts included. (D) Relative abundances of the top 20 most abundant genera without the endosymbionts included. Taxa are indicated by different colors in the figure legend and are displayed in descending order of relative abundance as indicated by their order in the legend. (E) Heatmaps showing the square root transformed relative abundances of the top 10 most abundant genera found among early nymphs, late nymphs and adult S. longipalpa.

At the genus level, with the endosymbionts, Blattabacterium was highly abundant (59.1% ± 5.2 SEM), followed by Wolbachia (24.7% ± 3.9 SEM), Akkermansia (2.2% ± 0.67 SEM), Desulfolvibrio (2.1% ± 0.57 SEM), Bacteroides (1.7% ± 0.43 SEM), Lactobacillus (1.6% ± 0.46 SEM), Alistipes (1.6% ± 0.32 SEM), and Parabacteroides (1.2% ± 0.36 SEM) (Fig. 1C). Without the endosymbionts, Lactobacillus was the most abundant (21.7% ± 5.8 SEM), followed by Akkermansia (13.5% ± 3.3 SEM), Desulfolvibrio (11.4% ± 1.9 SEM), Bacteroides (7.8% ± 1.2 SEM), Alistipes (7.7% ± 0.85 SEM), Parabacteroides (5.8% ± 0.93 SEM), Erysipelatoclostridium (2.5% ± 1.1 SEM), and Dysgonomonas (2.4% ± 0.32 SEM), among a few others (Fig. 1D). Results of the most abundant bacterial genera in each developmental stage are summarized in heatmaps in Fig. 1E.

Minor differences in gut microbiome diversity and composition between nymph and adult stages of S. longipalpa

Alpha diversity analyses of the S. longipalpa gut microbiome revealed some significant differences between stages. After removing the endosymbionts, no significant differences were found in the number of observed genera between the early nymphs, late nymphs and adults (Fig. 2A; Kruskal-Wallis test; P-value = 0.48). However, gut microbiome alpha diversity (Shannon Index) was significantly higher in the late nymphs compared to the early nymphs (Fig. 2B; Kruskal-Wallis with Dunn’s Test; P-value = 0.03). No significant differences in alpha diversity were detected between the adult males and females (Figs. S1A and S1B). While principal coordinate analysis (PCoA plot) based on weighted Bray-Curtis distances showed that the early nymphs may have more similar gut microbiome composition compared to the late nymphs and adults, a PERMANOVA test indicated no significant differences (Fig. 2C; R2 = 0.12; F-statistic = 1.48; P-value = 0.172). Similarly, PCoA based on unweighted Bray-Curtis distances showed no significant differences (Fig. S2A; PERMANOVA: R2 = 0.06; F-statistic = 0.69; P-value = 0.77) either with or without the symbionts and broken down by males and females (Figs. S2B and S2C). To further probe differences in gut microbiome beta diversity between the stages of development, we investigated the interindividual variation. Analysis of the distance from centroid in ordination space, which indicates how dispersed microbiomes are relative to the average distance in their group (Sharma et al., 2022), found that the adults have higher inter-individual variation in bacterial community composition compared to the early nymphs (Fig. 2D; Wilcoxon Rank Sum test; P-value = 0.02). Overall, alpha and beta diversity analyses showed some notable differences between the early nymphs and later developmental stages (i.e., late nymphs and adults), which were more similar to each other.

Figure 2 Minor differences in gut microbiome diversity and composition among the stages of development in S. longipalpa.

(A) Boxplots displaying no differences in the number of observed genera between stages of development. (B) Boxplots displaying significant differences in Shannon alpha diversity (Kruskal-Wallis test; P-value = 0.03). (C) PCoA based upon weighted Bray-Curtis distances (PERMANOVA: R2 = 0.12; F-statistic = 1.48; P-value = 0.172). (D) Distances from centroid show higher interindividual variation in gut microbiome composition between the early nymphs and adults (an asterisk (*) shows significance by Wilcoxon rank sum test; P-value = 0.02).

Akkermansia is significantly more abundant in early nymphs

Since minor differences were found in gut microbiome diversity and variation, we further analyzed what bacterial taxa may be driving these differences. Differential abundance analysis found that Akkermansia was the only genus to be significantly different among the developmental stages in S. longipalpa, showing it was much higher in abundance in the early nymph stages relative to the late nymph stages (Fig. 3A; ANOVA; F-value = 20.55; P-value = 1.14e-05). Analysis of the other most abundant genera including Lactobacillus, Desulfolvibrio and Bacteroides did not show any significant differences between the developmental stages (Figs. S3A–S3C). Considering that heatmaps displayed possible differences in the abundance of the endosymbionts in each developmental stage (particularly Wolbachia), we also conducted a differential analysis of these taxa. However, no statistical significance was detected (Fig. 3B; ANOVA; F-value = 0.038; P-value = 0.97 and Fig. 3C; ANOVA; F-value = 1.487; P-value = 0.25). Further analysis of differential abundances between adult males and females found only one statistically significant difference; the females displayed significantly higher abundances of Parabacteroides than males (Fig. S4B; Student’s t-test; P-value = 0.04). Analysis of the differential abundances of the other most abundant genera found in adult males and females can be seen in Fig. S4.

Figure 3 Akkermansia is significantly more abundant in the early nymph stages of S. longipalpa.

Boxplots illustrating the centered-log ratio (CLR) transformation of abundances. (A) Boxplots demonstrating Akkermansia is higher in abundance during the early nymph stages compared to the late nymph stages (an asterisk (*) indicates significance by one-way ANOVA; F-value = 20.55; P-value = 1.14e−05). (B) Boxplots demonstrating no significant difference in the abundance of the endosymbiont Blattabacterium. (C) Boxplots showing differential abundances of Wolbachia but no statistical significance reached.

Blattabacterium abundance is negatively correlated with microbiome alpha diversity and abundance of the secondary endosymbiont Wolbachia

To consider whether the highly abundant endosymbionts may affect each other or other constituents of the gut microbiota, we performed several correlations. Pearson’s correlation analysis performed with the ten most abundant bacterial taxa among all the developmental stages found Blattabacterium to be negatively correlated with them all. Blattabacterium and Wolbachia showed a particularly strong and statistically significant negative correlation (Fig. 4B; Pearson correlation coefficient r = −0.82; Multiple R2 = 0.68; P-value = 8.131e−07). Meanwhile, Wolbachia was positively correlated with all the other bacterial taxa except for Akkermansia and Parabacteroides (Fig. 4A). Furthermore, we also observed Blattabacterium to be strongly negatively correlated with gut microbiome alpha diversity (Fig. 4C; Pearson correlation coefficient r = −0.78; Multiple R2 = 0.62; P-value = 5.749e−06), while Wolbachia showed no correlation with gut microbiome alpha diversity (Fig. 4D; Pearson correlation coefficient r = 0.31; Multiple R2 = 0.09; P-value = 0.14).

Figure 4 Correlations between endosymbionts and the gut microbiome.

(A) Pearson’s correlation analysis performed with the top ten most abundant bacterial taxa detected in S. longipalpa. (B) Pearson’s correlation demonstrated a strong negative correlation between Blattabacterium and Wolbachia (r = −0.82; Multiple R2 = 0.68; P-value = 8.131e−07). (C) Pearson’s correlation revealed a strong negative correlation between Blattabacterium and gut microbiome alpha diversity (Shannon’s Index) (r = −0.78; Multiple R2 = 0.62; P-value = 5.749e−06). (D) No correlation was found between Wolbachia and gut microbiome alpha diversity (r = 0.31; Multiple R2 = 0.09; P-value = 0.14).

Discussion

Here, we present the first molecular study comprehensively characterizing the microbiota of the brown-banded cockroach (S. longipalpa), both with and without its endosymbionts, across developmental stages and sexes. Our results indicate that the microbial community in early nymphs is distinct from later stages, which suggests developmentally driven changes in the gut microbiome. In addition, out data suggest that primary Blattabacterium endosymbionts play an important role in shaping the total microbial community, as an increase in Blattabacterium relative abundance was specifically associated with lower gut microbiota diversity and lower relative abundance of the secondary endosymbiont Wolbachia.

It is well established that pest cockroaches, particularly B. germanica and the American cockroach, Periplaneta americana, harbor a wide variety of microorganisms in their gut that assist a number of important physiological processes. However, the microbial composition of S. longipalpa had not been previously elucidated. While we found that the Blattabacterium and Wolbachia endosymbionts make up over 80% of the microbial community composition in S. longipalpa, we also found some similarities to other cockroach species in the composition of the gut microbiome. For example, some of the most abundant bacterial taxa at the family level detected in B. germanica were also present in S. longipalpa, including Desulfovibrionaceae, Bacteroidaceae, Ruminococcaceae, and Rikenellaceae (Kakumanu et al., 2018; Pérez-Cobas et al., 2015). However, S. longipalpa also harbored Lactobacillaceae and Akkermansiaceae in higher abundances. At the genus level, microbial taxa discovered in S. longipalpa were also found in three species of Periplaneta cockroaches including Lactobacillus, Desulfovibrio, Dysgonomonas, Bacteroides, Tennerella, Parabacteroides and Alistipes (Lee et al., 2020).

Overall, the microbial communities in the early nymphs were observed to be different from the late nymphs and adults, which were more similar to each other. Microbial alpha diversity was found to be significantly lower in the early nymphs compared to the late nymphs, while lower inter-individual variation was also observed in early nymphs relative to adults. Fully deciphering diversity patterns in S. longipalpa may require future analysis of individual gut sections, but our observations are similar to developmental changes discovered in B. germanica (Carrasco et al., 2014). In ticks (Amblyomma americanum) (Menchaca et al., 2013) and several species of crustaceans (Mente et al., 2016; Miao et al., 2020; Zhang et al., 2021), changes in the microbiome during development have also been reported. Although our data do not address the mechanisms behind the patterns we describe, we speculate several possibilities. Developmental differences in the gut microbiome may be due to differences in feeding behavior that introduce bacteria into the gut (Pérez-Cobas et al., 2015; Zhu et al., 2022), including coprophagy, which is more prevalent in early instar nymphs (Cruden & Markovetz, 1987; Kakumanu et al., 2018; Pietri & Kakumanu, 2021). Additionally, biochemical changes in the gut microenvironment that occur as the gut matures may exert selection on the microbial community, inhibiting the survival of some taxa while promoting the growth of others, resulting in ecological succession.

Interestingly, a major difference observed in the gut microbiome of S. longipalpa across development was the greater relative abundance of Akkermansia in early nymphs. Some species in this genus (Akkermansia muciniphila) have been similarly found to colonize the gut early in life in humans (Derrien et al., 2008; Rodrigues et al., 2022), but Akkermansia has not been widely observed in the gut microbiome of cockroaches. To our knowledge, only one metagenomic study of the gut microbiome of B. germanica detected Akkermansia in both the early nymph and adult stages (Domínguez-Santos et al., 2021). Akkermansia spp. are Gram-negative anaerobic bacteria that can colonize the gut of a broad range of mammals (Belzer & de Vos, 2012). Although our study could not identify the species of Akkermansia present, different species, particularly A. muciniphila, have been identified as potential probiotic bacteria, similar to Lactobacillus spp. (Rodrigues et al., 2022), suggesting that both Akkermansia and Lactobacillus could be necessary for optimal fitness of S. longipalpa in the indoor environment.

An additional, particularly notable finding was the apparent relationship between the primary endosymbiont Blattabacterium and the secondary endosymbiont Wolbachia. Considering all life stages and sexes, there was a strong, statistically significant negative correlation between the relative abundances of the two endosymbionts, as higher abundance of Blattabacterium was linked to lower abundance of Wolbachia. This link was much stronger than the link between Blattabacterium and the levels of other abundant taxa such as Akkermansia and Lactobacillus. The biological significance of this correlation remains unclear, but it suggests competition between the two endosymbionts for intracellular space and/or resources, as both are known to occupy fat body and germline tissues to enable vertical transmission (Pietri, DeBruhl & Sullivan, 2016). As such, S. longipalpa may be a promising model for the study of dual endosymbiont interactions at both the organismal and cellular levels. A similar negative relationship was also observed between Blattbacterium and the gut microbiome, as higher relative abundance of this endosymbiont was specifically significantly negatively correlated with gut microbiome alpha diversity (Shannon Index), while the abundance of Wolbachia was not. In studies of experimentally generated aposymbiotic B. germanica, other investigators found that Blattabacterium and gut microbiota had distinct physiological roles and did not compensate for each other nor interact significantly (Cazzaniga et al., 2023; Muñoz-Benavent et al., 2021). Our data indicate that S. longipalpa differs in this regard. However, fully establishing these links will require targeted follow-up studies of the absolute abundances of both endosymbionts across development.

From a practical standpoint, the new insight into the microbiome of S. longipalpa presented here has several important implications. There is ample evidence that the gut microbiota plays important positive roles in the physiology of other cockroaches, including insecticide resistance (Chao et al., 2020; Jiang et al., 2021; Pietri, Tiffany & Liang, 2018; Wolfe & Scharf, 2022; Zhang et al., 2018, 2022; Zhang & Yang, 2019). While it remains to be determined if this is the case in S. longipalpa, understanding the composition of its gut microbiome will inform the design of studies to test the links between the microbiome and physiology and suggest potential bacterial taxa to target, either chemically or environmentally, to mitigate any microbe-mediated resistance or other microbe-mediated benefits. In addition, it is well known that cockroaches have the ability to acquire, maintain, and transmit human enteric bacterial pathogens (Donkor, 2020; Turner, Peta & Pietri, 2022). Although S. longipalpa has been found to harbor such pathogens in the field (Nasirian, 2016), little is known about the mechanistic aspects of pathogen transmission by this species. Recent work has revealed that the transmission of enteric bacterial pathogens such as Escherichia coli and S. Typhimurium by German cockroaches is a complex, biologically active process that is influenced in part by the gut microbiota (Ray, Potts & Pietri, 2020; Turner, Van Hulzen & Pietri, 2023; Turner, Peta & Pietri, 2022). Therefore, the microbiota may also be a potential target for mitigating pathogen transmission by S. longipalpa, and its role in the ability to become infected by and transmit pathogens is intriguing. Both of the above avenues should be investigated further, and the information provided by this study is a pivotal first step towards extending our knowledge of the microbial communities of S. longipalpa beyond characterization and into the functional and practical realms.

Supplemental Information

Supplemental Information 1 Supplementary Figures.

Additional Information and Declarations

Competing Interests

Author Contributions

Data Availability

The authors declare that they have no competing interests.

Kylene Guse conceived and designed the experiments, performed the experiments, analyzed the data, prepared figures and/or tables, authored or reviewed drafts of the article, and approved the final draft.

Jose Pietri conceived and designed the experiments, authored or reviewed drafts of the article, and approved the final draft.

The following information was supplied regarding data availability:

All raw sequencing data associated with this work is available at NCBI SRA: PRJNA1046101.

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
