# Peer review of "Endosymbiont and gut bacterial communities of the brown-banded cockroach, Supella longipalpa"

_PeerJ, doi:10.7717/peerj.17095_

## Round 0.1 · original submission · Major Revisions

Dear Authors,

Having carefully reviewed the feedback from three reviewers, it is with pleasure that I convey their unanimous consensus on the manuscript's commendable quality and its merit for publication. Nevertheless, valuable suggestions have been put forth concerning the experimental design. Additionally, it is a requirement of the journal that the obtained sequences be deposited in a repository such as the NCBI SRA database. Consequently, major corrections are advised.

Best Regards,

Armando Sunny

Reviewer 1 ·

Basic reporting

It is an interesting research that studied and quantified both the endosymbiont and gut bacterial communities of Supella longipalpa for the first time. And the mechanism of shaping overall community configuration was preliminarily elucidated. It is meaningful for extending knowledge of the microbial communities of S. longipalpa. This manuscript reflects a lot of work by the authors, but there are some inappropriate aspects in it.
In conclusion, this manuscript lacks some key information, and figures are unclear, so it needs to publish after major revisions. Please check and revise carefully, some suggestions are listed below, hoping to help the author improve the quality of the manuscript.

Experimental design

1.In “Materials and Methods”, “Cockroaches”, how much instar is determined as early or late instar? You should cite relevant references to describe it.
2.In “Materials and Methods”, “Data analysis”, the endosymbiont was removed, but did not describe the method of removing. Please describe the method of removing endosymbiont, and indicate that endosymbiont had been completely removed and gut communities were not affected.

Validity of the findings

1.The sequencing results should be submitted to the NCBI SRA database and provide the accession numbers to search.
2.The repeatability of the data in the heat map is relatively low in Fig 1A,B,C,D .
3.The experiment distinguished gender when selecting materials, and you distinguished gender in Fig 1E, but Fig 2,3 were not. Please distinguish gender in the Fig 2,3.
4.There are only five samples from early instar nymphs in Fig 2C, it should be six to be right, please correct it.

Additional comments

1.The format of the citation of references is not correct. For example, line 65, “(Li et al., 2020; Rosas et al., 2018; Wolfe and Scharf, 2021; Zha et al., 2023)”, you should follow chronological order when labeling references in the main text, you can correct to “(Zha et al., 2023; Wolfe and Scharf, 2021; Li et al., 2020; Rosas et al., 2018)”.
2.In “References”, some references are not labeled with page numbers, such as line 364,367, please carefully check and use the correct format.
3.Fig 1 is not clear, the color is obscure and figure legends cannot be recognized. Please improve the quality of figures.
4.The p-value needs to be in uppercase italics.
5.The symbol of significant difference in text is different from in figure.You should use the same symbol.
6.Line 286, “Salmonella Typhimurium” corrected to “Salmonella typhimurium”,
Line 324 also has the same issue.

Reviewer 2 ·

Basic reporting

no comment

Experimental design

no comment

Validity of the findings

no comment

Additional comments

The article by Guse and Pietri analyzed the microbiome of brown-banded cockroach, Supella longipalpa, an indoor cockroach species with the aim to quantitatively examine and characterize both its endosymbiont and gut bacterial communities. The authors performed 16SrRNA amplicon sequencing on 24 samples comprising early and late instar nymphs and male and females from a lab colony. The data shows a significant effect of nymphal development on the gut microbiome composition and also a significant negative correlation between the two endosymbionts Blattabacterium and Wolbachia. The authors have presented an interesting data showing the correlation between different microbial taxa between life stages of brown banded cockroaches. I have a few comments below for the authors to consider.

Comments:
Comment 1: The authors have done a nice job of analyzing the microbiome data to understand the dynamics of gut microbial communities and endosymbionts during different life stages of S. longipalpa. Using differential abundance analysis, they showed the abundant taxa in the potential gut communities after bioinformatically removing the endosymbiont reads which is very helpful. However, I have a few comments here:
1. Choice of DNA extraction method: The aim of the study is to characterize the gut and endosymbiotic bacteria which resides outside the gut in cockroaches. The DNA was extracted from whole insects. While I agree that the whole insect sampling is very useful to understand the composition of the whole microbiome, it completely lost the resolution into the gut microbiome given the abundance of endosymbionts (>80% of relative abundance). Why did authors choose not to include gut dissection samples separately which would give a clear gut microbial composition?
2. Part of the objective of the paper is also to quantitatively examine the microbial composition. I agree that the authors have done extensive analysis of microbiome data to measure the quantitative variations between samples. But the study is completely lacking the absolute abundance data. Since there are only a small number of samples, I suggest the authors do qPCR analysis to show the absolute abundance of total microbial load / absolute abundance of 1 or 2 individual taxa which will really help the reader to better relate the data.
3. Correlations of symbionts: It is very interesting to see that there is a negative correlation between the 2 symbionts. This could be very important biologically/physiologically. However, it is well known that the relative abundance may not represent the absolute abundance. Eg: If Blattabacterium concentration increases over time, since it is in relative abundance, it automatically decreases the abundance of other taxa. To support these observations from microbiome data I suggest the authors quantify (by qPCR) one or both the endosymbionts with proper controls and normalization.

Other comments:
Introduction: I commend the authors for providing detailed literature review of other common cockroach species in relation to S. longipalpa. While it is very helpful, I think there is very little information about Supella itself. I suggest authors to add more description of the S. longipalpa (eg: life cycle, distribution and its importance as pest species etc)
Methods: Line 96-97: Please provide some clarity on early instars: Is it first instar or second instar (or how many days after hatching etc). Also, please specify the age of males and females that were analyzed in the study?
Line 125 -142: Data Analysis: Please provide details about the normalization of the data while conducting all these analyses?
Please include the project accession number to the manuscript if there is any.
Results: Lines 146 - 159: Please provide the ±SEM for all the values provided in the parenthesis.
Fig 4: It would be helpful if Fig 4 images are modified to show both males and females instead of combining them as adults.
Discussion: Line 249: I believe the variation is more in the composition not in the diversity of microbial taxa. Please make it clear unless there are any microbial taxa that are uniquely present in the early instars which is driving this variation If so, please mention those taxa.
lines 273 - 291: This paragraph is confusing. It is not clear if the author is claiming Akkermansia to be beneficial or pathogenic to S. longipalpa. Since not much is known about this bacterium in cockroaches, I suggest to keep this discussion simple. Line 280-291 is adding more confusion to the paragraph.
Line 307-310: I cannot agree with this statement, unless it is shown by absolute abundance. Please see my comment 1-3 above.
Lines 286 and 291: capitalization of S. typhimurium

·

Basic reporting

This manuscript, especially the introductory part, is exceptionally well written. The author justified the need for the work in unambiguous terms.

Experimental design

I do not have any reservations about the approach sought in this work. The authors were interested in characterizing the bacterial communities of the brown-branded roach. Likewise, they appear interested in how developmental stages shape these communities. Their approach is scientifically backed.

However, I do have a comment on the experimental design. Admittedly, my comment is an observation, not a flaw of the experimental design.

Why did the authors not dissect the roaches (particularly adults) and separate the fore, mid, and hindgut? Perhaps, if the authors had done so, they would have been able to relate their findings of the diversity of microbial communities to localization. And by extension resolve or explain the Akkermansia pattern.

German cockroaches have 3-5 nymphs depending on environmental factors and nutrients. How many instars do the brown-banded roach have? What is the age of the ones included in this study? Could age affect the distribution of the microbial communities? If yes, authors should probably include the age of the different developmental stages used.

Validity of the findings

The data and interpretation thereof are adequate. The findings from this paper are quite informational and a necessary first step towards informing our knowledge of the microbial communities of the brown-banded roach.

However, figure 1 is very problematic and not easy on the eye. The fonts are quite small and a better quality should be uploaded. I struggled to make out the wording and interpret the data myself. Zooming the PNG images compromised resolution. Perhaps a higher dpi image should be used. I had to largely follow the lead of the authors. #All figures could probably benefit from an increase in font.

Questions/Comments
1. Line 164-167. Why weren't the adults analyzed separately since that was the convention in Fig1E?

2. Line 189-192. Data shows that Akkermansia is significantly more abundant in early nymphs. So, the authors argue that's probably due to molting (L255-264). I remain unconvinced and here is why.
- On the surface, molting should not reap the gut contents of an insect. This is because, during apolysis, there is a separation of the old cuticle from the underlying epidermis. It is this old cuticle that is digested and shed off. However, this is only a shallow explanation.

- To back their argument (molting removes fore and hindgut), the authors cite a couple of papers that were either not relevant or wrong. For example, Jahnes et al 2021 is a wrong reference. The right reference should be Takashima and Hartenstein 2012. Interestingly, Jahnes et al 2021 cited the work of Takashima and H 2012 to back the argument the authors in this manuscript used. Remarkably, Takashima and H's 2012 paper made no mention of the word "molting" in the entire manuscript. The authors would probably need to revise this section of the manuscript.

3. Line 228-229: "Our results indicate that the microbial community composition in early nymphs is distinct.." I am not sure I follow. Fig. 2 essentially shows a minor difference (as illustrated by Shannon's DI - a measure of species richness) which is more in late nymphs (not early nymphs). Or does instinct here implicate Akkermansia?

4. if yes, how are the early nymphs more distinct? The roaches were sourced from the same colony. Cockroaches acquire microbes via feeding (on exuviae, coprophagy, or diet). What developmental changes could have shaped the microbial community? Please do not say molting. Or better still provide apposite backings.

3. Why is Wolbachia more abundant in males? What evolutionary and cost-benefit role would this serve since Wolbachia is vertically transmitted?

Additional comments

1. What does microbial alpha diversity measure?
2. Remove italics L259.
3.

---

## Round 0.2 · accepted · Accept

Dear Authors,

It is with great pleasure that I share the exciting news of your manuscript's acceptance for publication. The reviewers' feedback has been overwhelmingly positive, affirming the readiness of your work for publication. We extend our sincere gratitude to you for choosing PeerJ as the platform to present your remarkable research.

Warm regards,

Armando Sunny

Reviewer 1 ·

Basic reporting

no comment

Experimental design

no comment

Validity of the findings

no comment

Additional comments

no comment

·

Basic reporting

The authors satisfactorily tended to queries and utilized suggestions where apposite.

Experimental design

NA

Validity of the findings

NA